# Have maternal or paternal ages any impact on the prenatal incidence of genomic copy number variants associated with fetal structural anomalies?

**Marta Larroya**[1]*, **Marta Tortajada**[1], **Eduard Mensión**[1], **Montse Pauta**[2], **Laia Rodriguez-Revenga**[2,3,4], **Antoni Borrell**[1]*

**1** BCNatal—Barcelona Center for Maternal-Fetal and Neonatal Medicine (Hospital Clínic and Hospital Sant Joan de Deu), Institut Clinic de Ginecologia, Obstetricia i Neonatologia, IDIBAPS, Fetal i+D Fetal Medicine Research and Centre for Biomedical Research on Rare Diseases (CIBER-ER), University of Barcelona, Barcelona, Catalonia, Spain, **2** Institut d'Investigacions Biomèdiques August Pi i Sunyer (IDIBAPS), Barcelona, Catalonia, Spain, **3** Biochemistry and Molecular Genetics Department, Hospital Clinic of Barcelona, Barcelona, Catalonia, Spain, **4** CIBER of Rare Diseases (CIBERER), Instituto de Salud Carlos III, Madrid, Spain

* larroya@clinic.cat (ML); aborrell@clinic.cat (AB)

**Data Availability Statement:** The datasets presented in this article are not readily available because qChip® Pre is a microarray design legally

## Abstract

The objective of this study was to determine whether maternal or paternal ages have any impact on the prenatal incidence of genomic copy number variants (CNV) in fetuses with structural anomalies. We conducted a non-paired case-control study (1:2 ratio) among pregnancies undergoing chromosomal microarray analysis (CMA) because of fetal ultrasound anomalies, from December 2012 to May 2020. Pregnancies with any pathogenic copy number variant (CNV), either microdeletion or microduplication, were defined as cases. Controls were selected as the next two pregnancies with the same indication for CMA but with a normal result. Logistic regression was used, adjusting by use of assisted reproductive technology (ART) and parental smoking. Stratified analysis was performed according to CNV type (*de novo*/inherited and recurrent/non-recurrent). The study included 189 pregnancies: 63 cases and 126 controls. Mean maternal age in cases was 33.1 (SD 4.6) years and 33.9 (SD 6.0) years in controls. Mean paternal mean age was 34.5 (SD 4.8) years in cases and 35.8 (SD 5.8) years in controls. No significant differences in maternal or paternal age were observed, neither in stratified analysis according to the CNV type. Moreover, the proportion of cases were not significantly different between non-advanced and advanced ages, either considering paternal or maternal ages. The presence of pathogenic CNV at CMA in fetuses with structural anomalies was not found to be associated with advanced paternal or maternal age.

protected by Quantitative Genomic Medicine Laboratories SI (Esplugues de Llobregat, Catalonia Spain). Requests to access the datasets should be directed to info@qgenomics.com or atencdb@clinic.cat.

**Funding:** The authors received no specific funding for this work.

**Competing interests:** The authors have declared that no competing interests exist.

## Introduction

Aneuploidies are the leading genetic cause of miscarriage and developmental disabilities in humans. Genetic recombination is known to contribute to human aneuploidies by producing nondisjunction of homologous chromosomes, although an understanding of the molecular mechanisms responsible for meiotic non-disjunction remains elusive. Advanced maternal age has been long recognized as the primary risk factor for nondisjunction. It is well established that the frequency of aneuploidies dramatically increase in older mothers [1, 2]. Regarding a paternal age effect, some studies have reported that advanced paternal age is associated with a significantly higher risk of fetal chromosomal defects [3], although these findings were not confirmed by other studies [4].

Advanced paternal age has been suggested to be a risk factor for an adverse pregnancy outcome (stillbirth, preterm birth and low birthweight), for structural anomalies (encephalocele and microcephalia, cleft lip and palate, neural tube defects and cardiac defects) and for some childhood disorders (autism spectrum disorder, schizophrenia or cancer) [5–10]. Regarding chromosomal anomalies, most of *de novo* structural aberrations are paternal in origin, in contrast to aneuploidies that are mainly of maternal origin. This may reflect a vulnerability of premeiotic divisions of germ cells to structural rearrangements, as there are many more of these divisions in the male than in the female germ cell line [11, 12]. Furthermore, some monogenic disorders have been reported to be more frequent in advanced paternal age, mainly skeletal dysplasias and craniosynostotic syndromes [13–15]. As a man ages, the number of *de novo* single nucleotide variant (SNV) in his sperm increases, and the chance that a child would carry a mutation that could lead to monogenic disorders increases proportionally [16].

There are scarce data on intermediate sized genomic imbalances, between 10 kb and 10 Mb, known as genomic copy number variations (CNV), also called submicroscopic chromosomal anomalies or microdeletions and microduplications. Although some studies have suggested an association between clinical neuropsychiatric disorders with both an increased paternal age and CNV [11, 17], no link was found between CNV and advanced paternal age in an extensive Dutch study including psychiatric, neurologic and urologic patients [17]. No studies have been reported on pathogenic CNVs in fetuses with ultrasonographically detected structural defects. Our study will explore if advanced parental ages are linked to prenatal incidence of genomic CNV in structurally abnormal fetuses.

## Materials and methods

We conducted a retrospective non-paired case-control study (1:2 ratio) in the Prenatal Diagnosis Unit of BCNatal-Hospital Clínic Barcelona including pregnancies in which chromosomal microarray analysis (CMA) was performed during an 8-year study period (December 2012 to May 2020) because of fetal structural anomalies or ultrasound markers. Cases were defined as those fetuses in which CMA revealed a submicroscopic anomaly, defined as a genomic imbalance < 10Mb, after exclusion of pregnancies achived by assisted reproductive technologies with gamete donation and those procedures performed because a family history. The variants of uncertain significance were excluded from the study. Controls were selected as the following two fetuses with the same main reason for CMA resulting in a normal result. The case list found during the study period determined the sample size.

Data were collected retrospectively by reviewing prenatal and obstetric records. Information on demographic data, smoking, singleton/twin pregnancy, utilization of assisted reproductive technology (ART), result of first trimester combined test, ultrasound findings including type of structural malformations, gestational age at diagnosis, type of invasive prenatal diagnosis and maternal and paternal age at conception were obtained. The results of

invasive prenatal diagnostics were retrieved from our department of genetics. The case list of genetic microdeletions was merged with our prenatal database.

Maternal age was obtained as the exact age at conception directly from our database. Biological paternity was not assessed. Paternal age was obtained as the age reached at conception through a telephone interview with the mother. The father provided informed oral consent to have data from their medical records used in this study, given that the need for written consent was waived by the ethics committee. Advanced maternal age was defined as ≥35 years while advanced paternal age was defined at ≥40 age according to previous studies [18]. The first trimester combined test includes maternal age at the expected date of delivery, nuchal translucency measurement, serum free beta-human chorionic gonadotrophin (β-hCG) and pregnancy-associated plasma protein A (PAPP-A). A high-risk result was defined when at least one of the two estimated risks, for trisomy 21, and for trisomy 18–13, was above 1:250.

CMA is a technology used for the detection of CNV, either microdeletions or microduplications, as well as conventional chromosomal anomalies. It is able to detect genomic changes as small as 10Kb in size—a resolution up to 1000 times higher than that of conventional karyotyping.

CMA is used for uncovering CNVs thought to play an important role in the pathogenesis of a variety of disorders, primarily congenital anomalies. Reasons for CMA were divided into eight different categories, most of them types of structural fetal anomalies: neurologic, cardiovascular, nephro-urologic, skeletal, multiple malformations, in addition to ultrasound aneuploidy markers, fetal growth restriction (FGR) and other anomalies such as polyhydramnios and stillbirth. FGR was defined by the statistical deviation of an estimated fetal weight from a population-based reference, below the third centile, or alternatively to fetuses with an EFW below the 10th centile and a concomitant abnormal fetal or uterine Doppler.

During the first four years (2012–2015) of the study, a BAC (bacterial artificial chromosome) microarray (CytoChip Focus Constitutional, BlueGenome, Illumina, San Diego, California, USA) was used with a 1 Mb resolution along the whole genome and 100 kb in 143 constitutional regions associated with pathology. In the four last years of the study (2016–2020) a high-resolution oligonucleotide microarray (qChip Prenatal from qGenomics, Barcelona, Catalonia, Spain) was used. The qChip Prenatal microarray is based on an Agilent 8x60K format and provides a high coverage of clinically relevant regions (1 probe/10Kb on average). In all cases, scanning and image acquisition were carried out using an Agilent microarray scanner (G2565BA) and data analysis was performed using the BlueFuse Multi software (BlueGnome) or the qGenviewer software (qGenomics) as reported elsewhere [19].

The Kolmogorov-Smirnov test and visual plot inspection were used to assess the normality of continuous data distributions. We performed a correction in paternal age, adding 0.5 years in each male individual to correct the bias of truncated age. Categorical data are presented as n (%) and continuous data as mean [standard deviation (SD)] or median [interquartile range (IQR)] according to their distribution. A Kruskal-Wallis test and a two-tailed Mann–Whitney U test were used to compare differences of arithmetic variables between the groups. Comparisons of proportions were performed using Chi-square or Fisher's exact tests. Student's t-test or the Mann-Whitney U-test and Pearson Chi-square test were used to perform univariate comparisons between groups of quantitative and qualitative variables, respectively. Power was calculated for a case-control study with a sample size of 189 and 1:2 ratio, assuming a relevant difference of 2 years between groups, with a result of 75.6%. The sample size was determined by the cohort of fetuses that met the inclusion criteria during the study period (December 2012 to May 2020), the first 8 years of CMA application in our center. This is the complete cohort of pathogenic CNVs found at CMA in chromosomally abnormal fetuses studied in our center. The power was calculated according to our sample size. Differences on the proportion

of cases between non- advanced and advanced age was assessed for maternal and paternal ages.

We performed a logistic regression to assess the impact of maternal and paternal age in the CMA result, adjusting by possible confounders as utilization of ART and maternal and paternal smoking. We have chosen these two possible confounders as there is some evidence that ART and smoke could increase the risk of genetic anomalies in oocytes and sperm [20, 21]. Furthermore, stratified analysis was performed in recurrent and non-recurrent CNVs and *de novo* and inherited CNVs. Some CNVs are recurrently observed in the human genome, and they are thought to arise by homologous recombination between repeated sequences, a process called non-allelic homologous recombination. Missing data were assumed to miss at random. For all analyses, a two-tailed p value <0.05 was considered significant. All the statistical analyses were performed using Stata/IC 15.1©.

## Results

During the 8-year study period (December 2012 to May 2020), 63 structurally abnormal fetuses with a pathogenic CNV were enrolled as cases after exclusion of 9 pregnancies achived by assisted reproductive technologies with gamete donation and 21 procedures performed because a family history and 126 controls were selected as the following two fetuses with the same structural anomaly and a normal CMA result. 21 variants of uncertain significance were excluded from the study. The mean gestational age at the diagnosis of fetal anomalies was 23.1 (SD 7.4) weeks. Prenatal diagnosis was performed by amniocentesis in 150 (79%) pregnancies and by chorionic villus sampling in 39 (21%) pregnancies. There were no significant differences between cases and controls in maternal and pregnancy characteristics (Table 1).

The most frequent indication for CMA was cardio-vascular defects (32%) followed by neurologic anomalies (16%). Table 2 shows the frequencies for the eight different categories. Among the 63 submicroscopic pathogenic anomalies, 15 microduplications and 48 microdeletions were found. There were 28 (44%) recurrent and 7 (11%) inherited CNVs (S1 Table).

**Table 1. Maternal and pregnancy characteristics.**

| | CONTROL GROUP (N = 126) | CASE GROUP (N = 63) | P-VALUE |
|---|---|---|---|
| MATERNAL ETHNICITY | Caucasian 97 (77.0%) | Caucasian 52 (82.5%) | 0.542 |
| | Latin American 10 (7.9%) | Latin American 8 (12.7%) | |
| | Asian 10 (7.9%) | Asian 1 (1.6%) | |
| | Magreb 7 (5.6%) | Magreb 2 (3.2%) | |
| | African 2 (1.6%) | African 0 (0%) | |
| SMOKER (CIG/DAY) | Mother 17 (13.8%) | Mother 10 (16.1%) | 0.677 |
| | Father 30 (28.0%) | Father 18 (29.0%) | 0.891 |
| MATERNAL BMI† (KG/M²) | 24.1 [4.4] | 22.9 [3.1] | 0.062 |
| NULLIPAROUS | 50 (40.3%) | 32 (50.8%) | 0.174 |
| PREVIOUS AFFECTED | 3 (2.4%) | 2 (3.1%) | 0.771 |
| ASSISTED REPRODUCTIVE TECHNOLOGIES | 17 (13.5%) | 9 (14.3%) | 0.882 |
| MULTIPLE PREGNANCY | 10 (7.9%) | 5 (7.9%) | 1.000 |
| FIRST TRIMESTER COMBINED TEST | Low risk 86 (68.8%) | Low risk 41 (65.0%) | 0.915 |
| | High risk 20 (16.0%) | High risk 14 (22.2%) | |
| | Unknown 20 (16.0%) | Unknown 8 (12.7%) | |

Categorical data are presented as "n (%)" and continuous data as "mean [standard deviation (SD)]".

†BMI: *Body Mass Index*

**Table 2.  Type of anomalies diagnosed by ultrasound.**

| Type of anomaly | Cases | Controls |
|---|---|---|
| | N = 63 | N = 126 |
| Cardio-vascular | 20 (31.8) | 40 (31.8) |
| Neurologic | 10 (15.9) | 20 (15.9) |
| Aneuploidy markers | 7 (11.1) | 14 (11.1) |
| Multiple | 7 (11.1) | 14 (11.1) |
| Nephro-urologic | 7 (11.1) | 14 (11.1) |
| Fetal Growth Restriction | 6 (9.5) | 12 (9.5) |
| Skeletal | 2 (3.2) | 4 (3.2) |
| Other | 4 (6.3) | 8 (6.3) |

Categorical data are presented as "n (%)".

Mean maternal age in cases was 33.1 (SD 4.6) years and 33.9 (SD 6.0) years in controls. Mean paternal mean age was 34.5 (SD 4.8) years in cases and 35.8 (SD 5.8) years in controls (Fig 1). There were no significant differences in maternal or paternal age in non-adjusted logistic regression, and even after adjusting for possible confounders such as ART and maternal or paternal smoking (Table 3). Stratified analysis in recurrent and non-recurrent CNVs,

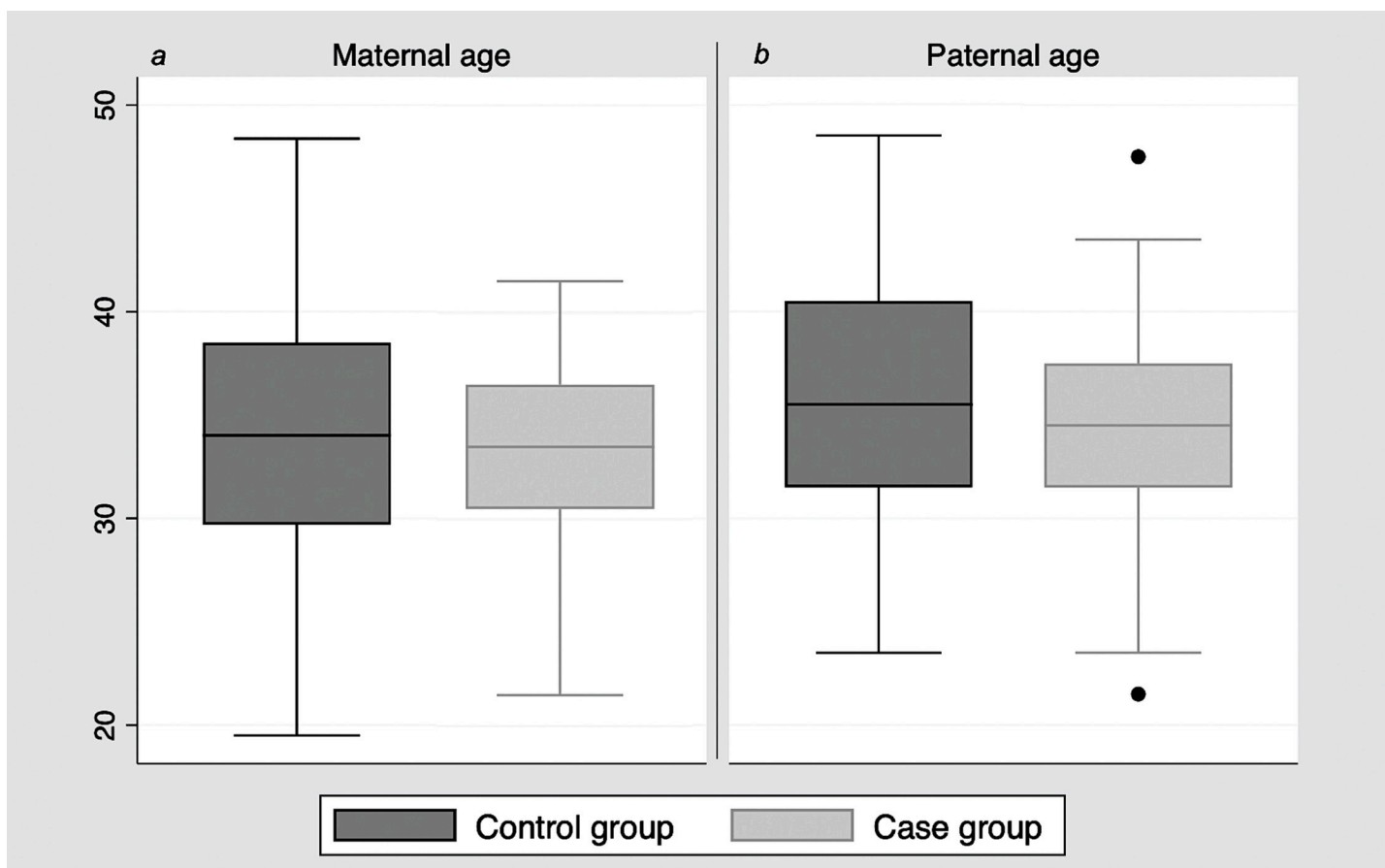

**Fig 1. Comparison of maternal and paternal age between cases and controls.** Box plots represent the spread of maternal age (a) and paternal age (b) among controls (dark gray, left, absence of CNVs) and cases (light gray, right, presence of CNVs).

**Table 3. Comparison of maternal and paternal ages between cases and controls.**

|  | MATERNAL AGE | | PATERNAL AGE | | |
| --- | --- | --- | --- | --- | --- |
|  | *OR* | *95% CI* | *OR* | *95% CI* | *p-value* |
| Non-adjusted | 1.059 | 0.954–1.175 | 0.913 | 0.822–1.015 | *0.179* |
| Adjusted | 1.078 | 0.963–1.206 | 0.894 | 0.800–1.001 | *0.363* |

Adjusted values by assisted reproductive technologies and maternal and paternal smoking. OR: Odds Ratio. 95% CI: 95% Confidence Interval.

**Table 4. Comparison of maternal and paternal age between cases and controls stratified according recurrence and inheritance type.**

|  | MATERNAL AGE | | PATERNAL AGE | | |
| --- | --- | --- | --- | --- | --- |
|  | *OR* | *95% CI* | *OR* | *95% CI* | *p-value* |
| Recurrent CNV | 1.049 | 0.901–1.220 | 0.906 | 0.780–1.052 | *0.597* |
| Non- recurrent CNV | 1.109 | 0.964–1.277 | 0.880 | 0.766–1.011 | *0.533* |
| Inherited CNV | 1.185 | 0.874–1.605 | 0.828 | 0.614–1.117 | *0.799* |
| *De novo* CNV | 1.066 | 0.950–1.195 | 0.903 | 0.807–1.010 | *0.451* |

OR: Odds Ratio. 95% CI: 95% Confidence Interval.

**Table 5. Comparison between the proportion of cases in advanced and non-advanced ages according to the maternal and the paternal age.**

|  | Maternal Age | | Paternal Age | |
| --- | --- | --- | --- | --- |
|  | < 35 years | ≥ 35 years | < 40 years | ≥ 40 years |
| Total | 112 | 77 | 144 | 45 |
| Cases | 41 | 22 | 53 | 10 |
| Proportion of cases | 0.366 | 0.286 | 0.368 | 0.222 |
| p-value | 0.275 | | 0.100 | |

and also in *de novo* and inherited CNVs resulted in no significant differences in any of the categories (Table 4). Results were assumed as non-statistical significance since all OR 95% confidence intervals included number 1. Moreover, no differences were observed in the proportion of cases between groups of non-advanced and advanced age either for maternal and paternal ages (Table 5).

## Discussion

### Main findings

Our study in fetuses undergoing CMA for ultrasound anomalies demonstrated that there were no differences in either paternal or maternal age between fetuses affected by pathogenic CNVs as compared with those with a normal CMA result. Maternal age effect in the origin of fetal aneuploidy and a paternal age effect in *de novo* SNV particularly those related to skeletal anomalies and neurodevelopmental disorders are well established, but the effect of parental age seems not to be relevant in the origin of CNV, either recurrent or non-recurrent.

### Comparison with the literature

An increased risk for genomic CNV has been reported in intellectual disability, neurodevelopmental disorders and congenital defects [22]. In the prenatal field, fetal structural defects

including also increased nuchal translucency [23], FGR [24] and stillbirth [25] have been shown to carry an increased risk for pathogenic CNVs. However, advanced maternal age has not been shown to carry and incremental risk for CNV above pregnancies of the general population [26].

Data from postnatal studies have shown that most of the CNV are of paternal origin. A Dutch study including 118 subjects with intellectual disability and nonrecurrent *de novo* CNV found that 76% of CNVs in this cohort originated in the paternal allele. This paternal bias was independent of CNV characteristics: type (loss/gain), genomic size (<1 Mb or >1 Mb) and genomic mechanism involved (recurrent or non-recurrent). Recurrent *de novo* CNVs are often flanked by segmental duplications in the direct vicinity of the CNV breakpoints that mediate the generation of these rearrangements through non-allelic homologous recombination. Interestingly enough, a significant increase in paternal age was observed in the 74 cases of *de novo* CNVs without segmental duplications in the direct vicinity of their breakpoints. This results suggest that increased paternal age may have a major impact on the generation of non-recurrent CNVs [17]. A similar British study including 173 patients with physical and/or neurological abnormalities and a de novo imbalance identified by array CGH found more paternally derived imbalances. The majority of those imbalances arose through male-specific mechanisms other than non-allelic homologous recombination, such as mitotic mechanisms, although a paternal-age effect was not observed [12].

A large prospective population-based cohort study in the Netherlands, including 6,501 healthy participants with parental age information, found no evidence of paternal age effect on CNV load in the offspring. In contrast to studies that suggest *de novo* single nucleotide variant (SNV) rate to be dominated by paternal age at conception, these results strongly suggest that at the level of global CNV burden there is no influence of increased paternal age. While it remains possible that local genomic effects may exist for specific phenotypes, this study supports that global CNV burden and increased father's age may be independent disease risk factors [27].

The effect of paternal age on the rate of *de novo* SNV and CNVs in healthy subjects, was explored in selected ten families consisting of two monozygotic twins and their parents from the Quebec Study of Newborn Twins (QSNT) project [28]. This study demonstrated a strong positive correlation between paternal age and germline *de novo* SNV in healthy subjects using disease-free familial quartets. However, germline CNVs did not follow the same pattern, because the CNV rate did not change with parental age. This finding supports the current evidence that CNV and single nucleotide variations (SNV) have different mechanism, given that it is thought that most CNVs in the human genome arise from non-allelic homologous recombination, while a vast majority of the SNVs occur during DNA replication.

There is a single prenatal series in which the paternal age was recently studied in fetuses carrying a genomic CNV. This large series from a laboratory from United States, included 127 prenatal cases, in which samples were collected according to clinical indications for prenatal diagnosis, reported non-significant differences between cases (*de novo* CNVs) and controls (inherited CNVs). However, in the postnatal series, a maternal age effect was observed to be associated with a higher rate of *de novo* recurrent CNV [29].

## Clinical application

Advanced maternal age was the most important reason for diagnostic amniocentesis during many decades. Nowadays advanced paternal age, arbitrarily defined as 40 years or more, is suggested to be a reason to screen for the most common autosomal dominant monogenic disorders. Meanwhile, the incidence of CNVs does not appear to be influenced by either maternal or paternal age.

Our study is the first to explore the link between paternal age and CNV in congenital birth defects, particularly fetal structural anomalies detected by prenatal ultrasound. Although CNVs account for about 1% of liveborn fetuses, similar to the chromosomal anomaly rate, only a small proportion of liveborn with a pathogenic CNV can be prenatally suspected by a previous history or fetal ultrasound anomalies. This is an argument to support the universal offering of CMA to all pregnant women [30].

## Strengthens and weaknesses

Our study is not without limitations. A major shortcoming of the study is the fact that the parent of origin of their CNVs, especially their de novo CNVs, was not studied. By using a population that is limited to fetuses with abnormal ultrasounds, there is already an increased likelihood of an underlying genetic cause. If no pathogenic CNV is found, it is possible that the ultrasound anomalies exist because of an SNV or other underlying mutation not able to be ascertained by the CMA platform. As advanced paternal age is considered a risk factor for SNVs, the "control" group with normal CMAs may have an increased paternal age because these are actually fetuses with de novo SNVs. Another limitation of the present study is that it was conducted in a single-center that acts as a tertiary referral center, where we receive fetal structural anomalies from different cities of Catalonia and Spain. Extremely advanced maternal age or utilization of ART can be one of the reasons for referral, resulting in an older pregnant women population. Furthermore, as this is one of the first studies exploring the link between paternal age and CNV, we could overlook possible confounders currently unknown. In addition, despite the fact that this is a substantive and commendable number of as clinical case studies go in amassing fetal structural anomalies with array CMA data for each case, it is still is a limited number of cases to examine the paternal age effect. On contrary, a strength of our study is that consecutive cases during the study period were included, avoiding selection biases. We conducted a case-control study in which both groups shared the same reason for CMA. This is relevant because it can lead to explore our hypothesis in different groups of fetal structural anomalies.

## Supporting information

**S1 Table. Description of the 63 pathogenic copy number variants included in the study using the International System for Human Cytogenetic Nomenclature (ISCN) and divided into recurrent and non-recurrent.**
(DOCX)

## Author Contributions

**Conceptualization:** Marta Larroya, Montse Pauta, Antoni Borrell.

**Data curation:** Marta Larroya, Marta Tortajada, Laia Rodriguez-Revenga.

**Formal analysis:** Marta Larroya.

**Methodology:** Marta Larroya, Eduard Mensión, Laia Rodriguez-Revenga, Antoni Borrell.

**Supervision:** Antoni Borrell.

**Validation:** Laia Rodriguez-Revenga, Antoni Borrell.

**Writing – original draft:** Marta Larroya.

**Writing – review & editing:** Antoni Borrell.

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
