## [Decision Letter · Decision Letter 0]

1 Apr 2021

PONE-D-21-01707

Have maternal or paternal ages any impact on the prenatal incidence of genomic copy number variants associated with fetal structural anomalies?

PLOS ONE

Dear Dr. Larroya,

Thank you for submitting your manuscript to PLOS ONE. After careful consideration, we feel that it has merit but does not fully meet PLOS ONE’s publication criteria as it currently stands. Therefore, we invite you to submit a revised version of the manuscript that addresses the points raised during the review process.

We look forward to receiving your revised manuscript.

Kind regards,

Rogelio Cruz-Martinez, Ph.D.

Academic Editor

PLOS ONE

Journal Requirements:

https://jmg.bmj.com/content/48/11/776

https://link.springer.com/article/10.1007/s00439-012-1261-4

https://journals.plos.org/plosone/article?id=10.1371%2Fjournal.pone.0164212

In your revision ensure you cite all your sources (including your own works), and quote or rephrase any duplicated text outside the methods section.

Further consideration is dependent on these concerns being addressed.

3. We note that you are reporting an analysis of a microarray, next-generation sequencing, or deep sequencing data set. PLOS requires that authors comply with field-specific standards for preparation, recording, and deposition of data in repositories appropriate to their field. Please upload these data to a stable, public repository (such as ArrayExpress, Gene Expression Omnibus (GEO), DNA Data Bank of Japan (DDBJ), NCBI GenBank, NCBI Sequence Read Archive, or EMBL Nucleotide Sequence Database (ENA)). In your revised cover letter, please provide the relevant accession numbers that may be used to access these data. For a full list of recommended repositories, see http://journals.plos.org/plosone/s/data-availability#loc-omics or http://journals.plos.org/plosone/s/data-availability#loc-sequencing

4. Please provide additional details regarding participant consent.

In the ethics statement in the Methods and online submission information, please ensure that you have specified (i) whether consent was informed and (ii) what type you obtained (for instance, written or verbal, and if verbal, how it was documented and witnessed). If your study included minors, state whether you obtained consent from parents or guardians. If the need for consent was waived by the ethics committee, please include this information.

Additional Editor Comments:

The Abstract’s conclusion is not supported by the results in this section. The authors should include the definition of advanced parental age, and to assess the proportion of CNV between the group with and without advanced maternal and paternal ages. Please include the number of cases with CNV of unknown significance.

Reviewers' comments:

Reviewer's Responses to Questions

**Comments to the Author**

1. Is the manuscript technically sound, and do the data support the conclusions?

Reviewer #1: Partly

Reviewer #2: Yes

2. Has the statistical analysis been performed appropriately and rigorously? 

Reviewer #1: No

Reviewer #2: Yes

3. Have the authors made all data underlying the findings in their manuscript fully available?

Reviewer #1: Yes

Reviewer #2: Yes

4. Is the manuscript presented in an intelligible fashion and written in standard English?

Reviewer #1: Yes

Reviewer #2: Yes

5. Review Comments to the Author

Reviewer #1: Thank you for allowing me to review this study. In this manuscript, the authors examined the effect of maternal and paternal age on the incidence of copy number variants detected by microarray in fetuses with anomalies on ultrasound. The study is a matched retrospective case-control study matching in a 1:2 ratio and included 63 cases and 129 controls. The hypothesis was that parental age would be higher in fetuses with CNVs. Overall, however, the authors show that maternal and paternal age are no different in fetuses with CNVs compared to controls. If anything, paternal age was borderline lower in cases.

This study is novel, in that a similar study has not been done in fetuses. However, as acknowledged by the authors, there is already considerable postnatal data from a cohort of almost 6000 patients suggesting that parental age is not very relevant for risk of CNV, unless in some specific conditions. As such, the results of the present study are not very surprising and do not add new scientific knowledge.

Overall however, the manuscript is well written, clear and easy to follow. The methodology seems correct.

I do have a few more concerns:

- It is a bit atypical to see power calculated at 75.6%. Typically, one would aim for 80% or higher. Was this a post-hoc power analysis? I wonder why such a small sample size was used. Only 63 cases were included in 8 years. What was the reason for this low number? Especially in a retrospective analysis, much higher numbers should be available. Why were some cases excluded?

- How did the authors ascertain paternity? Some of the assumed fathers may not have been the biological fathers of the pregnancy.

- Were families with known parental CNVs excluded? Indeed, inclusion of families with known CNVs could bias the results of the findings.

Reviewer #2: The article is a case-control study performed to analyze if maternal and paternal age have an impact on the prenatal incidence of CNVs.

1.Advanced paternal age is also a risk factor for other structural anomalies such as: midline anomalies (neural tube defects, cardiopathy and cleft lip and palate) (Jemal et al and Hook).

Authors should include this sentence and reference.

2.-The objective of the study was to determine whether the paternal or maternal age have an impact on the incidence of CNVs. It is important to break down maternal and paternal age by groups and perform a statistical analysis of each group instead of only comparing mean maternal and paternal ages.

6. PLOS authors have the option to publish the peer review history of their article (what does this mean?). If published, this will include your full peer review and any attached files.

Reviewer #1: No

Reviewer #2: **Yes: **Monica Aguinaga

---

## [Author Response · Author response to Decision Letter 0]

13 May 2021

Thank you for the opportunity to review our manuscript. We have addressed the Journal requirements and Reviewers’ comments:

Journal Requirements:

Done. 

 2. We noticed you have some minor occurrence of overlapping text with the following previous publication(s), which needs to be addressed https://jmg.bmj.com/content/48/11/776

https://link.springer.com/article/10.1007/s00439-012-1261-4

https://journals.plos.org/plosone/article?id=10.1371%2Fjournal.pone.0164212 In your revision ensure you cite all your sources (including your own works), and quote or rephrase any duplicated text outside the methods section. Further consideration is dependent on these concerns being addressed.

Duplicated texts in the Discussion have been already solved

 3. We note that you are reporting an analysis of a microarray, next-generation sequencing, or deep sequencing data set. PLOS requires that authors comply with field-specific standards for preparation, recording, and deposition of data in repositories appropriate to their field. Please upload these data to a stable, public repository (such as ArrayExpress, Gene Expression Omnibus (GEO), DNA Data Bank of Japan (DDBJ), NCBI GenBank, NCBI Sequence Read Archive, or EMBL Nucleotide Sequence Database (ENA)). In your revised cover letter, please provide the relevant accession numbers that may be used to access these data. For a full list of recommended repositories, see http://journals.plos.org/plosone/s/data-availability#loc-omics or http://journals.plos.org/plosone/s/data-availability#loc-sequencing

We have tried to deposit microarray data in the ArrayExpress public repository. However since we are using a microarray design that has not been previously uploaded and is legally protected we are afraid that we could not achieve this query. Since we are a diagnostic laboratory with a certified quality management system (ISO 9001 AENOR, https://www.en.aenor.com/certificacion/calidad) that routinely perform external quality control assessment (European Molecular Genetics Quality Network (EMQN) and the Genomics Quality Assessment (GenQA)) for all tests, including prenatal and postnatal microarray analysis, we wonder if could include a “Data availability Statement” indicating that that all data is available upon request without any restriction.

4. Please provide additional details regarding participant consent. In the ethics statement in the Methods and online submission information, please ensure that you have specified (i) whether consent was informed and (ii) what type you obtained (for instance, written or verbal, and if verbal, how it was documented and witnessed). If your study included minors, state whether you obtained consent from parents or guardians. If the need for consent was waived by the ethics committee, please include this information.

Accordingly, a new sentence was added in the Methods section: “The father provided informed oral consent to have data from their medical records used in this study, given that the need for written consent was waived by the ethics committee”.

Additional Editor Comments:

The Abstract’s conclusion is not supported by the results in this section. 

The Abstract has been rewritten “The presence of pathogenic CNV at CMA in fetuses with structural anomalies was not found to be associated with an advanced paternal or maternal age.”

The authors should include the definition of advanced parental age, and to assess the proportion of CNV between the group with and without advanced maternal and paternal ages. 

We have included the definition of advanced maternal and paternal age according to previous studies: “Advanced maternal age defined as ≥ 35 years while advanced paternal age was defined at ≥ 40 years according to previous studies (Elise de La Rochebrochard, Patrick Thonneau, Paternal age and maternal age are risk factors for miscarriage; results of a multicentre European study, Human Reproduction, Volume 17, Issue 6, June 2002, Pages 1649–1656)”.

Furthermore, we conducted a new analysis including the proportion of CNV between groups with and without advanced maternal and paternal ages. We have added new text and a new table as follows: 

Abstract: “Moreover, the proportion of cases were not significantly different between non-advanced and advanced age, either considering paternal or maternal ages.”

Methods: “Differences on the proportion of cases between non- advanced and advanced age was assessed for maternal and paternal ages.”

Results: “Moreover, no differences were observed in the proportion of cases between groups of non-advanced and advanced age either for maternal and paternal ages (Table 5)”.

Table 5. Comparison between the proportion of cases in advanced and non-advanced ages according to the maternal and the paternal age. 

 Maternal Age Paternal Age

 < 35 years ≥ 35 years < 40 years ≥ 40 years

Total 112 77 144 45

Cases 41 22 53 10

Proportion of cases 0.366 0.286 0.368 0.222

p-value 0.275 0.100

Please include the number of cases with CNV of unknown significance.

We have add the following sentence in Methods: “The variants of uncertain significance were excluded from the study.” And in Results: “The 21 variants of uncertain significance were excluded from the study.” 

Reviewers' comments:

Reviewer's Responses to Questions

Comments to the Author

1. Is the manuscript technically sound, and do the data support the conclusions?

Reviewer #1: Partly

Reviewer #2: Yes

2. Has the statistical analysis been performed appropriately and rigorously? 

Reviewer #1: No

Reviewer #2: Yes

3. Have the authors made all data underlying the findings in their manuscript fully available?

Reviewer #1: Yes

Reviewer #2: Yes

4. Is the manuscript presented in an intelligible fashion and written in standard English?

Reviewer #1: Yes

Reviewer #2: Yes

5. Review Comments to the Author

Reviewer #1: Thank you for allowing me to review this study. In this manuscript, the authors examined the effect of maternal and paternal age on the incidence of copy number variants detected by microarray in fetuses with anomalies on ultrasound. The study is a matched retrospective case-control study matching in a 1:2 ratio and included 63 cases and 129 controls. The hypothesis was that parental age would be higher in fetuses with CNVs. Overall, however, the authors show that maternal and paternal age are no different in fetuses with CNVs compared to controls. If anything, paternal age was borderline lower in cases.

This study is novel, in that a similar study has not been done in fetuses. However, as acknowledged by the authors, there is already considerable postnatal data from a cohort of almost 6000 patients suggesting that parental age is not very relevant for risk of CNV, unless in some specific conditions. As such, the results of the present study are not very surprising and do not add new scientific knowledge.

Overall however, the manuscript is well written, clear and easy to follow. The methodology seems correct.

Thanks for these comments

I do have a few more concerns:

- It is a bit atypical to see power calculated at 75.6%. Typically, one would aim for 80% or higher. Was this a post-hoc power analysis? I wonder why such a small sample size was used. 

We have included this text on Material and Methods: “The sample size was determined by the cohort of fetuses that met the inclusion criteria during the study period (December 2012 to May 2020), the firsts 8 years of CMA application in our center. The power was calculated according to our sample size.” 

Only 63 cases were included in 8 years. What was the reason for this low number? Especially in a retrospective analysis, much higher numbers should be available. Why were some cases excluded?

This is the complete cohort of pathogenic CNVs found at CMA in structurally abnormal fetuses studied in our center after exclusion of pregnancies achieved by assisted reproductive technologies with gamete donation and those procedures performed because a family history. One sentence was included in the Methods and another in the Results section.

- How did the authors ascertain paternity? Some of the assumed fathers may not have been the biological fathers of the pregnancy.

We have added a new sentence “Biological paternity was not assessed.” And the following sentenced was changed to better explain how we obtained the paternal age. “Paternal age was obtained as the age reached at conception through a telephone interview with the mother.” 

- Were families with known parental CNVs excluded? Indeed, inclusion of families with known CNVs could bias the results of the findings

Families with known parental CNVs were excluded, as stated now in the Methods, but those unknown at the time of sampling and subsequently found to be inherited were not. To avoid this bias, a stratified analysis was performed and de novo and inherited cases were assessed separately. 

Reviewer #2: The article is a case-control study performed to analyze if maternal and paternal age have an impact on the prenatal incidence of CNVs.

1.Advanced paternal age is also a risk factor for other structural anomalies such as: midline anomalies (neural tube defects, cardiopathy and cleft lip and palate) (Jemal et al and Hook). Authors should include this sentence and reference.

These structural anomalies have now been added in the Introduction. Two new references have been included: 

Ernest B. Hook Laurence E. Karp. Genetic counseling dilemmas: Down syndrome, paternal age, and recurrence risk after remarriage. First published: 1980 https://doi-org.sire.ub.edu/10.1002/ajmg.1320050206

Janeczko D, Hołowczuk M, Orzeł A, Klatka B, Semczuk A. Paternal age is affected by genetic abnormalities, perinatal complications and mental health of the offspring. Biomed Rep. 2020 Mar;12(3):83-88. doi: 10.3892/br.2019.1266. Epub 2019 Dec 20. PMID: 32042416; PMCID: PMC7006092.

2.-The objective of the study was to determine whether the paternal or maternal age have an impact on the incidence of CNVs. It is important to break down maternal and paternal age by groups and perform a statistical analysis of each group instead of only comparing mean maternal and paternal ages.

As stated before, we have conducted a new analysis including the proportion of CNV between groups with and without advanced maternal and paternal ages. We have added new text and a new table as follows: 

Furthermore, we conducted a new analysis including the proportion of CNV between groups with and without advanced maternal and paternal ages. We have added new text and a new table as follows: 

Abstract: “Moreover, the proportion of cases were not significantly different between non-advanced and advanced age, either considering paternal or maternal ages.”

Methods: “Differences on the proportion of cases between non- advanced and advanced age was assessed for maternal and paternal ages.”

Results: “Moreover, no differences were observed in the proportion of cases between groups of non-advanced and advanced age either for maternal and paternal ages (Table 5)”.

Table 5. Comparison between the proportion of cases in advanced and non-advanced ages according to the maternal and the paternal age. 

 Maternal Age Paternal Age

 < 35 years ≥ 35 years < 40 years ≥ 40 years

Total 112 77 144 45

Cases 41 22 53 10

Proportion of cases 0.366 0.286 0.368 0.222

p-value 0.275 0.100

---

## [Editor Report · Decision Letter 1]

15 Jun 2021

Have maternal or paternal ages any impact on the prenatal incidence of genomic copy number variants associated with fetal structural anomalies?

PONE-D-21-01707R1

Dear Dr. Larroya,

We’re pleased to inform you that your manuscript has been judged scientifically suitable for publication and will be formally accepted for publication once it meets all outstanding technical requirements.

Kind regards,

Rogelio Cruz-Martinez, Ph.D.

Academic Editor

PLOS ONE

---

## [Editor Report · Acceptance letter]

30 Jun 2021

PONE-D-21-01707R1 

Have maternal or paternal ages any impact on the prenatal incidence of genomic copy number variants associated with fetal structural anomalies? 

Dear Dr. Larroya:

I'm pleased to inform you that your manuscript has been deemed suitable for publication in PLOS ONE. Congratulations! Your manuscript is now with our production department. 

Kind regards, 

on behalf of

Dr Rogelio Cruz-Martinez 

Academic Editor

PLOS ONE